# The impact of different standing positions on gluteus medius activation and lumbar lordosis in LBP-developers during prolonged standing

**Saeedeh Abbasi**[1], **Hooman Minoonejad**[1]*, **Hamed Abbasi**[2], **Seyed Hamed Mousavi**[1]

**1** Department of Sports Injury and Biomechanics, Faculty of Sport Sciences and Health, University of Tehran, Tehran, Iran, **2** Department of Sport Injuries and Corrective Exercises, Sport Sciences Research Institute, Tehran, Iran

* h.minoonejad@ut.ac.ir

## Abstract

Low back pain (LBP) development has been associated with increased hip muscle co-activation and lumbar lordosis during standing in previously asymptomatic individuals. It is commonly advised to use footrests to relieve LBP. The impact of adjusting arm position on lumbar biomechanics can also be impressive. This study aimed to compare the effects of normalized footrest height and changing arm position on Gluteus medius (GMed) muscle activity, lumbar lordosis, and pain intensity. Twenty-four female pain developers (PDs) were recruited, identified by a >10 mm increase on the visual analog scale (VAS) during prolonged standing. Electromyography (EMG) recorded GMed activity, and photogrammetry measured lumbar lordosis at time points over one hour-standing. The first group (A) used the footrest intermittently, while the second group (B) additionally changed their arm positions. These variables were analyzed using repeated measures (between/within) ANOVA. No significant interaction was observed between the groups in right and left GMed co-contraction index (CCI) (p = 0.14). However, both groups exhibited a significant decrease in CCI during prolonged standing (time * condition: p = 0.003). Additionally, Group B consistently demonstrated lower overall levels of co-contraction across time (p = 0.01). An approximate 6-degree reduction in lumbar lordosis was observed after prolonged standing with both interventions (group A and group B; p = 0.008 and p = 0.01, respectively), although no significant differences in lumbar lordosis were detected between the groups. Lumbar discomfort increased over time; however, the interventions significantly alleviated this discomfort after the intervention time point. Notably, group B reported lower pain intensity compared to group A (p = 0.007). Applying these interventions in the workplace could be beneficial to reduce discomfort for individuals who stand for long periods of time. Further research is needed to optimize these strategies and assess long-term benefits.

## Introduction

Low Back Pain (LBP) is a prevalent and disabling condition worldwide, especially among individuals under the age of 45 [1,2]. The condition's prevalence varies due to multiple risk

**Data availability statement:** All relevant data are within the manuscript and its Supporting information files.

**Funding:** The author(s) received no specific funding for this work.

**Competing interests:** The authors have declared that no competing interests exist.

factors, including body measurements, limited spinal mobility, lumbar lordosis, and psychological influences [3,4]. A history of LBP significantly increases susceptibility to recurrent episodes. Notably, prolonged standing (typically resulting in discomfort after 30 to 45 minutes [3,5]) is identified as a risk factor for LBP, even in individuals without a prior clinical history of the condition [6]. This form of LBP often affects younger adults, particularly those aged 18 to 35 [7–9].

The etiology of LBP involves complex interactions between various factors, including anthropometric, structural, psychological, postural, and muscular aspects [6,10–17]. Reports suggest that individuals prone to standing-induced LBP, or Pain Developers (PDs), frequently exhibit biomechanical and motor control deficits. For instance, PDs demonstrate lower Gluteus Medius (GMed) endurance during side-bridge tests, often accompanied by reduced strength [5,18,19]. The bilateral co-contraction of GMed muscle plays a critical role in maintaining pelvic and lumbar stability during standing. Impaired or altered GMed activation has been linked to LBP in PDs, contributing to changes in motor control and muscle co-ontraction [20,21]. It is believed that this co-contraction compensates for the inability to properly control the trunk by the core muscles for postural stability [12,22,23]. The active hip abduction (AHAbd) test is often used to identify these deficits, as it predicts standing-induced LBP with high accuracy (0.83) [21,24,25]. Furthermore, task-specific factors, such as upper limb activity during standing, significantly influence GMed co-contraction, with implications for LBP development. Tasks requiring upper limb movement, like sorting or assembly, change GMed co-contraction, potentially reducing trunk instability and LBP risks [26]. Inadequate or inconsistent GMed activation, especially during dynamic tasks, may increase vulnerability to LBP in PDs. Lumbar lordosis is another pivotal factor in LBP development. PDs generally exhibit greater lumbar lordosis compared to non-PDs, with higher lordosis severity correlating with increased LBP symptoms [27,28]. One study noted a significant correlation with maximal visual analog scale (VAS) scores [28]. This association underscores the interplay between spinal posture and mechanical stress during prolonged standing. Research further links lumbar lordosis severity to negative sagittal vertical axis (SVA) displacement, indicating that postural deviations may exacerbate LBP symptoms [28–30]. Gender differences further complicate the understanding of LBP in PDs. Women generally exhibit greater lumbar flexion (a smaller anterior-posterior height ratio of intervertebral discs) compared to men. Women also tend to have higher endurance but reduced force capacity, suggesting gender-specific contributions to LBP susceptibility and motor control adaptations [31,32].

One potential intervention to improve lumbar alignment during prolonged standing involves optimizing arm positioning, which during standing can also influence sagittal spinal balance. Flexed shoulder positions or placing hands on the clavicles have been shown to improve lumbar lordosis and alleviate discomfort during prolonged standing [33–38]. Proper upper extremity and arm positions during standing activate muscles crucial for pelvic and lumbar stability, leading to improved lumbar lordosis and sagittal vertical axis. This may reduce co-contraction of trunk and pelvic muscles, or improve lumbopelvic alignment, alleviating pain and preventing back injuries from prolonged standing [36,39–41].

Assistive devices, such as steps and slope surfaces have also been studied for their biomechanical effects on the trunk and pelvic regions [17,22,42]. Various studies have investigated the impact of footrests on LBP during prolonged standing. Fatigue from prolonged standing typically develops after 48 to 120 minutes, with interventions implemented every 5 to 25 minutes [22,42,43]. The results suggest that an appropriate footrest height (10 to 13 cm or 10% of height) and regular foot switching can reduce pain and fatigue by increasing lumbar flexion and altering muscle activation patterns [3,22,24,42,44]. In contrast, improper prolonged use of footrests (5 or 15 minutes standing on footrest) may not yield positive effects [45]. Short-term

studies (not prolonged standing) have also shown that footrests can increase lumbar flexion, but they do not significantly affect muscle co-contraction patterns [22,44]. Therefore, adjusting footrest height and usage duration is crucial for achieving better outcomes [46,47]. A footrest height adjusted to achieve a 135-degree trunk-to-thigh angle has been shown to reduce lumbar load, fatigue, and pain intensity [42,44]. These devices, combined with optimized upper limb positions, may improve lumbopelvic alignment and alleviate discomfort during prolonged standing.

Since lumbar lordosis plays a pivotal role in the development of LBP, and prior research highlights the significant impact of arm positioning on lordosis, the lack of studies addressing this relationship in PDs constitutes a notable research gap. Moreover, footrest interventions have been shown to effectively reduce the activity of the GMed and lumbar muscles in PDs. This study investigates the combined effects of arm positioning and footrest usage during one hour of standing to improve lumbar-pelvic stability, alleviate pain intensity, and offer practical, generalizable solutions for reducing LBP risks in occupations requiring prolonged standing.

## Methods

### Participants

The sample size was determined using G*Power software version 3.1. With a confidence level of 0.95, an effect size of 0.83 [31], and a test power of 0.85, the minimum required sample size was calculated to be 8 participants. However, considering that previous studies included at least 12 participants, the test groups were set to 12 participants each. A total of 24 females (age: 29.15 (SD:3.68) years, height: 164.35 (SD: 4.76) cm; weight: 60.14 (SD: 8.50) kg; BMI: 22.28 (SD: 3.26) kg/m²)(with no history of back, shoulder, or arm pain (requiring medical intervention or more than three days off work in the past six months), previous back or hip surgery, inability to stand, and any dizziness or fainting during standing volunteered to participate in this study. Also, if they achieved scores above 13 in the Baecke physical activity questionnaire [48] (for the evaluation of habitual physical activity), they were excluded because our study did not include athletes with well-developed muscles. (Baecke Q scores: 9.36 (SD: 1.80)). Also, participants were excluded if they had employment involving standing in one place for more than 1 hour per day in the past 12 months, were unable to stand for more than 4 hours, had a body mass index (BMI) greater than 30, or reported any symptoms of LBP at the start of the standing task [28,49].

In the initial step, all participants underwent a 2-hour prolonged standing test to categorize them as either PD or NPD. A participant was classified as PD if their back pain VAS score exceeded 10 mm on a 100 mm scale during the standing test [19,50,51]; they often experienced their first pain at 20.85 (SD:10.21) minutes. The AHAbd test, designed to evaluate an individual's ability to maintain pelvic and trunk alignment while moving the lower limb in an unstable position, was also performed (S1 Table) [13,21,52]. All participants were identified as right-leg dominant. Leg dominance was established by asking participants to simulate four tasks: kicking a ball, stomping out a simulated fire, tracing a shape, and picking up a marble [53]. The AHAbd test results for the right and left foot were 2.2 (SD: 0.69) and 1.85 (SD: 0.36), respectively. The study received ethical approval from the Research Ethics Committee of Tehran University (R.UT.SPORT.REC.1402.038). Furthermore, the study is registered with the Iranian Registry of Clinical Trials (IRCT) under registration reference IRCT20230628058610N1. The recruitment period for this study began on 23 July 2023 and ended on 30 September 2023. Informed consent was obtained from participants for their involvement in the study and for the publication of identifiable information or images in an

open-access online journal. During data collection, access to personally identifiable information was restricted based on this consent, and all data were anonymized for analysis.

## Experimental protocol

On a separate day, participants stood for one hour in a confined workspace marked on the floor (0.50 m × 0.46 m) while completing a series of four 15-minute tasks. These tasks, selected to simulate basic occupational activities commonly performed during prolonged standing, included puzzle assembly, small object construction, form completion, and typing. The task order was randomized to reduce order effects.

A 20-cm footrest was used during the standing period. This height was selected to represent 10% of the participants' average body height and to ensure a 135° trunk-to-thigh angle [42,44] for the majority of participants, which ranged between 17 and 20 cm. After standing on level ground for ten minutes, participants placed their right leg on the footrest for one minute, returned to level standing for three minutes, and then repeated this process with their left leg. This five-minute protocol was repeated every ten minutes throughout the 60-minute standing period. Participants were randomly assigned to each group using "a computer-generated randomization list": Group A (intermittent footrest use; Fig 1A) and Group B (intermittent footrest use with arm flexion, hands crossed on the clavicles; Fig 1B).

## Instruments

Two pairs of wireless Myon 320 Surface EMG electrodes (Myon AG, Switzerland) [54] were placed over the right and left GMed muscles. The electrode was a SKINTACT® CT-601 type (Ag/AgCl differential electrode) that was placed halfway between the most superior aspect of the iliac crest and the greater trochanter [55,56]. EMG signals were collected at a frequency of 2000 Hz using a 16-bit A/D conversion card. The signal was differentially amplified using a common mode rejection ratio of 80 dB at 60 Hz and band-pass filtered from 20 to 450 Hz [47].

Maximum voluntary contraction (MVC) values were measured by instructing participants to perform maximal isometric contractions against resistance. To normalize and compare muscle activity relative to each muscle's maximal strength, MVCs were recorded for the left and right GMed. Participants performed hip abduction in a side-lying position, with resistance applied at the ankle by the experimenter, to obtain MVCs [55,56]. The experimenter can actively monitor and adjust the resistance applied, ensuring participants are performing maximal efforts without

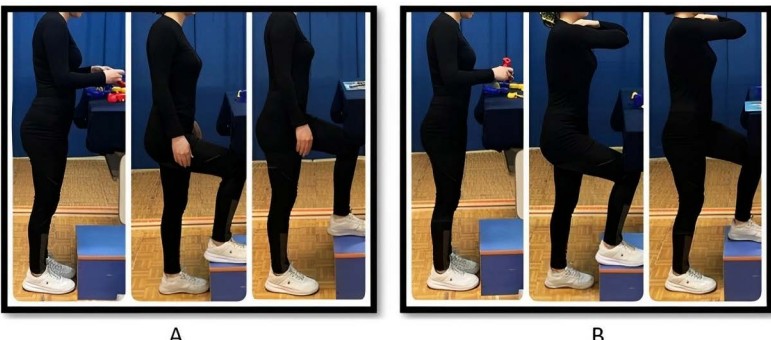

**Fig 1. Example of a participant using the elevated surface (20 cm) during the intermittent footrest use (A) and the intermittent footrest use with arm flexion, hands crossed on the clavicles (B) conditions.**

exceeding their limits, which could lead to injury. Each leg underwent two sets of MVCs, with a minimum of 30 seconds of rest between efforts. Additionally, 10-second resting trials were conducted with participants lying in both the supine and prone positions [3].

The lumbar lordosis angle was measured using photogrammetry with a Panasonic digital camera. Three infrared markers, or rigid bodies, were placed at the L1 (upper lumbar spine), L3, and L5/S1 levels [28,46]. Additionally, the VAS ruler was utilized to assess pain intensity.

## Data analysis

**Electromyography (EMG).** The sampling rate for surface electromyography (SEMG) signal recording was set at 2000 Hz, utilizing a 20–450 Hz band-pass filter [47]. The power frequency spectra of the raw data were initially analyzed using the fast Fourier transform (FFT) analysis pipeline in ProEMG 2.0 to identify characteristics of artifact noise. Raw EMG data were imported directly into a custom MATLAB (V.4) script for further processing, which included computing the time-domain and frequency-domain characteristics of the muscle activation. A Butterworth low-pass filter (450 Hz) and a high-pass filter (20 Hz) were applied before implementing a notch filter at 200 Hz. Double passes were filtered through a fourth-order Butterworth filter with a cutoff frequency of 6 Hz [20]. Each EMG linear envelope was normalized as a percentage of the maximum voluntary contraction (MVC). Subsequently, the root mean square (RMS) of the filtered raw data was calculated to determine the average intensity of the signal. This was achieved by averaging the squared values using the formula $1/N \sum x_i^2$. The normalized muscle activation data were expressed as a percentage of MVC using the formula (RMS EMG/ MVC RMS) * 100. This normalization facilitated comparisons across participants and conditions [31].

For each five-minute trial, data were segmented into three one-minute windows representing the following conditions: "Condition 1: before stepping," "Condition 2: right foot on the footrest with/without arm position change," and "Condition 3: left foot on the footrest with/without arm position change." These conditions were repeated four times after every 10 minutes of usual standing, over the course of one hour. Muscle activity was recorded during these three one-minute conditions.

The co-contraction index (CCI) was calculated to quantify the level of co-contraction between all possible muscle pairs (left and right GMed) using the following equation [57]:

$$CCI = \sum (EMG\ low/EMG\ high)\ (EMG\ low + EMG\ high)$$

This technique is commonly employed to evaluate the coordination between two muscles. Positive CCI values indicate concurrent activation of the muscles, while negative values suggest that one muscle is activated while the other is not, indicating muscle cross-firing [58].

**Kinematic data.** Using Kinovea software, after static calibration, three markers were selected to measure the relative angle of lumbar lordosis at any instance. The primary marker was placed at L3, with additional markers on L1 and L5/S1. For each participant, the relative angles of lordosis were recorded before and after the one-hour standing protocol.

**Visual analogue scale.** Low back discomfort was assessed using a 100-mm VAS before and after the stepping tasks with or without arm position changes, resulting in a total of eight scores. The minimum clinically significant difference for the VAS score was established at 10 mm, in accordance with previous research methodologies [13,45].

## Statistical analysis

All statistical tests were conducted using SPSS (Version 24.0) for Windows 10 (SPSS, Inc., Chicago, IL, USA). Dependent variables of interest include the percentage of maximum

excitation (%MVC) for EMG, the normalized change in joint angles for kinematic measures, and discomfort, which was measured by VAS displacement. A one-sample Shapiro-Wilk test was used to evaluate the normality of the distribution of variables. All variables showed a normal distribution. Independent variables included one between factors (groups) and two within factors (i.e., three levels of standing conditions and four levels of time). These variables were analyzed using repeated measures (between/within) ANOVA procedures and required additional comparisons. In situations where the data did not satisfy the assumption of sphericity, Greenhouse-Geisser corrected values were used that appropriately adjusted degrees of freedom for the statistical test.

In the co-contraction model; conditions (level ground standing, right foot on the footrest with/without change arm position, left foot on the footrest with/without change arm position), time (four repeated measurements over 1 hour) and the interaction effect of conditions and time (conditions * time) were modeled as independent variables within and between subjects for two groups.

In the VAS model, it was modeled in the same way, only the conditions were defined in two states before and after right foot stepping with/without change arm position.

A paired T-test was used to compare the value of the lumbar lordosis angle at the beginning and end of the test. The level of significance was set at α < 0.05 for all statistical tests.

## Result

### EMG (co-contraction)

Main effects and interactions, including time, condition, and the time * condition interaction, were observed in the analysis. While no significant interaction was found between the groups (p = 0.14), a significant within-group difference in LGMed–RGMed CCI was identified across the three conditions (p = 0.002). Additionally, significant effects were noted for time (p = 0.016) and the time * condition interaction (p = 0.006). When analyzed independently, Group A, which utilized a footrest intervention, exhibited a significant decreasing trend in CCI over time. Similarly, Group B, which incorporated both footrest use and changes in arm position, demonstrated a comparable trend across both time and conditions (Table 1). As illustrated in Fig 2, both groups displayed a similar pattern in the bilateral GMed muscles' CCI during one hour of standing, with Group B consistently showing lower co-contraction values than Group A.

### Kinematic

There was a non-significant effect on lumbar spine angle between both groups (p > 0.05). As shown in Table 2, a significant difference in lumbar lordosis was observed in both group between before and after the one-hour standing test.

**Table 1. Mean (95% CI) and standard deviation of Co-Contraction index of right and left gluteus medius (MVC%) during 60 minutes standing between two groups.**

| Group | Position 1: Standing on a flat surface | Position 2: Placing the dominant foot on the footrest with/ with-out changing the arm position | Position 3: Placing the non- domi-nant foot on the footrest with/ with-out changing the arm position | Test of within- sub-jects Effects in time | | Test of within-subjects Effects in condition | |
|---|---|---|---|---|---|---|---|
| | | | | (p-value) | F | (p-value) | F |
| A (Footrest) | 4.28 (3.73, 4.83) ± 1.90 | 3.20 (2.65, 3.38) ± 1.25 | 2.53 (2.31, 2.75) ± 0.77 | 0.41 | 1.01 | 0.003* | 10.55 |
| B (Footrest & Arm position) | 3.23 (2.80,3.66) ± 1.47 | 1.91 (1.68, 2.13) ± 0.76 | 2.01 (1.79, 2.24) ± 0.76 | 0.01* | 4.26 | 0.03* | 7.85 |

*p < 0.05.

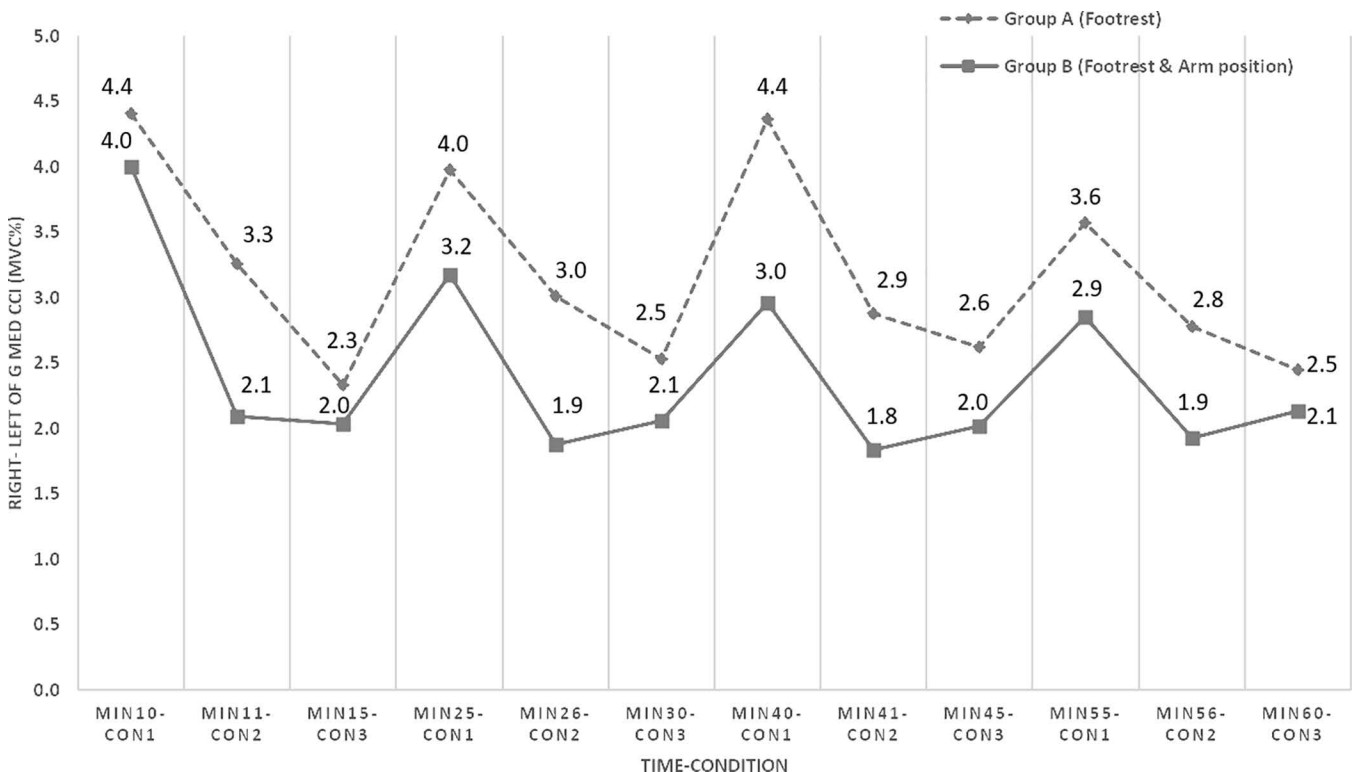

**Fig 2. Co-Contraction index of right and left gluteus medius activity (MVC%) during 60 minutes standing (time- condition) between two groups (A: used the footrest intermittently, B: used the footrest with change arm positions).** Con1:level standing, con2:right leg on footrest with/without changing arm position, con3: left leg on footrest with/without changing arm position.

**Table 2. Mean (95% CI) and standard deviation of Lumbar Lordosis angle(˚) between two groups pre and post 60 minutes of standing.**

| Group | Before 1- hour Standing | After 1-hour Standing | Test of within-subjects (p-value) |
|---|---|---|---|
| A (footrest) | 24.47 (20.12, 27.91) ± 5.41 | 18.62 (16.09, 21.15) ± 3.97 | 0.008 [*] |
| B (footrest & Arm position) | 23.44 (20.26, 26.60) ± 4.98 | 18.60 (16.80, 20.39) ± 2.82 | 0.01 [*] |

[*]$p < 0.05$.

## Discomfort (VAS)

A significant main effect of time (p = 0.001) was observed for lumbar discomfort in both groups, along with significant effects of condition (p = 0.001) and the time * condition inter-action (p = 0.03). On average, discomfort levels increased by 25.85 mm over time, regardless of the standing condition. However, after using the footrest—whether or not accompanied by changes in arm position—both groups demonstrated a significant decrease in discomfort over time and across conditions (Table 3). No significant difference in discomfort levels was observed between the two groups (p = 0.14), as shown in Fig 3.

## Discussion

The study highlights the efficacy of biomechanical interventions, specifically the intermittent use of a 20-cm footrest and strategic arm position adjustments, in alleviating LBP during

**Table 3. Mean (95% CI) and standard deviation of VAS Score (mm) before & after intervention between two group during 1-houre standing. Intervention: using footrest with/without changing arm position.**

| Group | Before intervention | After intervention | Test of within- subjects Effects in time | | Test of within- subjects Effects in condition | |
|---|---|---|---|---|---|---|
| | | | (p-value) | F | (p-value) | F |
| A (Footrest) | 17.26 (14.55, 19.97) ± 9.34 | 15.80 (13.39, 18.21) ± 8.30 | 0.00 * | 128.49 | 0.017 * | 11.44 |
| B (Footrest & Arm position) | 16.39 (14.17, 18.60) ± 7.64 | 13.91 (11.90, 15.92) ± 6.91 | 0.00 * | 36.07 | 0.007 * | 25.61 |

*p < 0.05.

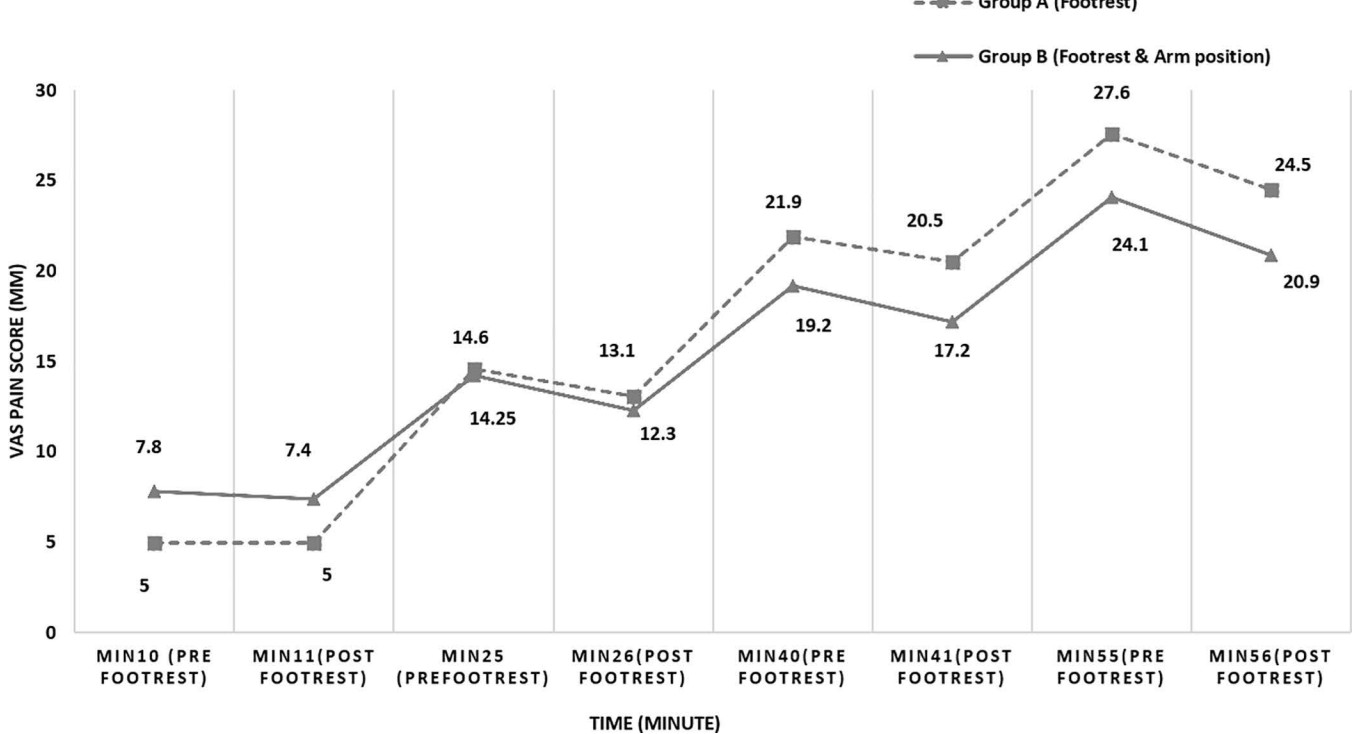

**Fig 3. VAS score (mm) before & after intervention between two group (A: used the footrest intermittently, B: used the footrest with change arm positions) every 15 minutes during 60 minutes of standing.**

prolonged standing. The findings indicate that while the overall interaction between groups (footrest use alone versus combined with arm position changes) was not statistically significant, the within-group differences underscore the differential impact of these interventions on lumbar lordosis, pain, and muscle co-activation patterns over time.

The arm position alteration, involving shoulder flexion with hands crossed on the clavicles influence upper body muscle activation, indirectly affecting lumbar alignment through kinetic chain interactions. Given that the weight of the arms increases the torque on the spine when moving forward and requires increased back muscle strength to maintain balance, this increase in muscle strength affects the fluctuation of body posture and consequently changes lumbo-pelvic parameters, including the amount of activity of the trunk muscles in stabilizing the back during standing [33]. Also, using footrest can cause slight lumbar flexion, reduced

lumbar torque, and decreased lumbar erector spinae muscle activity [42]. This aligns with prior findings by Sorensen et al. (2015) [28], which highlighted that an increase in lumbar lordosis correlates with elevated pain levels, particularly in individuals with postural dysfunction. The results also corroborate earlier work showing that specific arm positions, such as 90-degree shoulder flexion with hands placed on the clavicles, tend to minimize lordosis changes compared to more passive shoulder flexion with support or hands on the chest [33]. Interestingly, changes in arm position during standing also influenced SVA displacement. Positive SVA displacement—where the plumb line moves anterior to the sacrum—was reduced by 24% when participants placed their hands on their clavicles, compared to shoulder flexion alone [36]. Avotta et al. (2019) [38] similarly found a reduction in negative SVA displacement with the hands-on-clavicles position versus 45-degree shoulder flexion. These results suggest that modifying arm position may help participants maintain a flatter lumbar curve, thus promoting a more stable spinal balance during prolonged standing. The lumbar lordosis angle in this study decreased by approximately 6 degrees in both group. This decrease is consistent with findings from Fewster et al. (2017) [22], who reported enhanced lumbar spine flexion during intermittent foot elevation. In contrast, Callaghan and colleagues (2017) [8] did not observe significant differences in lumbosacral angles with footrest use alone, underscoring the value of arm positioning as a complementary strategy in reducing lumbar strain.

Our findings also underscore the role of GMed co-contraction as a potential compensatory mechanism for trunk stabilization. Increased co-contraction of GMed muscles during standing has been associated with elevated LBP, particularly among those with PD [12,22,23]. This study observed a reduction in GMed co-contraction within both groups. These findings suggest that footrest use and arm position adjustments optimize biomechanical alignment, alleviate lumbar strain, and enhance postural stability during prolonged standing. Moreover, engaging the upper extremities through core muscle activation can reduce GMed's compensatory role in trunk control and postural stability [12,22,23]. This reduction in GMed muscle strain during intermittent foot elevation and changing arm position may serve to alleviate some of the discomfort LBP. Consistent with Wong et al. (2010), implementing a training protocol to enhance trunk stability and increase rest time for the gluteus medius muscles during the early stages of standing led to a reduction in simultaneous contraction of these muscles [31]. Additionally, Wong et al. (2017) found that tasks involving upper limb immobilization tended to reduce co-contraction levels compared to more active tasks [44]. Regarding the use of footrests, Lee et al. (2018) [45] suggested that prolonged foot elevation might not effectively alleviate LBP due to excessive pressure on the standing foot. They hypothesized that biomechanical factors, rather than muscular fatigue, contribute to standing-induced pain and emphasized the importance of intervention details, such as footrest height and pain localization. Moreover, research indicates that spinal flexion increases with step height, which may exacerbate lumbar strain if weight-bearing on the standing foot is not properly optimized [47]. Sen et al. (2017) [42] demonstrated that using a footrest at 10% of body height, with 15-minute intervals, reduced LBP in individuals with PD. In a related study, Fewster et al. (2017) noted a significant reduction in pain with footrest use every 8 minutes during 80 minutes of standing. However, Foley et al. (2021) [47] reported no significant differences in LBP intensity across various footrest heights, highlighting the importance of optimizing footrest parameters for effective pain management. The trend of decreasing VAS scores over time suggests that step incorporation may help manage discomfort by minimizing external lumbar moments. Moreover, the consistently lower pain intensity reported by the arm position group emphasizes the additive effect of arm adjustments on LBP alleviation. It is conceivable that altering arm positioning helps decrease lumbar extension demands, allowing for reduced lumbar muscle activation and alleviating muscle fatigue [38,59]. This adjustment not only contributes

to pain relief but also underscores the role of strategic muscle activation as a stabilizing mechanism for prolonged standing and reducing the risk of lumbar instability and LBP.

These interventions hold promise for individuals required to stand for long periods, especially in occupational settings. They are practical, cost-effective, require no specialized equipment, and help prevent LBP while improving spinal health. While the study provides compelling preliminary evidence, several methodological limitations need to be considered. One of the main limitations is the small sample size, which limits the generalizability of the findings. A larger, more diverse sample would allow for better representation of various populations, particularly those with occupational standing requirements. Moreover, the lack of long-term follow-up limits the ability to assess the sustained effects of the interventions. Future studies should include longitudinal designs to examine whether the benefits observed in this study can be maintained over time and how they influence long-term musculoskeletal health.

## Conclusion

This study demonstrates that stepping, both alone and with added arm positioning, can significantly influence muscle co-contraction, lumbar spine posture, and discomfort levels during prolonged standing. Although no significant interaction between the two groups was found, the within-group analyses reveal meaningful differences and benefits associated with these interventions. Specifically, the footrest plus change arm position intervention showed a trend toward greater reductions in muscle co-contraction and discomfort, suggesting that it might be a more effective strategy for mitigating the negative effects of prolonged standing and prevent LBP. These findings have practical implications for designing workplace interventions to reduce musculoskeletal strain and improve comfort for individuals required to stand for extended periods in various occupational settings. One potential avenue is improving the work environment—such as adjusting workstation height, providing adjustable footrests, or ensuring proper ergonomic setups—could offer further relief by promoting better spinal alignment and reducing muscle fatigue. Future research should explore the long-term effects of these interventions and investigate other potential combinations of more movements and positions to further enhance their effectiveness.

## Supporting information

**S1 Table. Scoring criteria of Active Hip Abduction Test (AHAbd).**
(DOCX)

**S2 Table. Data.**
(XLSX)

**S3 Table. Raw data.**
(XLSX)

## Author contributions

**Conceptualization:** Saeedeh Abbasi, Hooman Minoonejad.

**Data curation:** Saeedeh Abbasi, Hooman Minoonejad, Seyed Hamed Mousavi.

**Formal analysis:** Saeedeh Abbasi, Hooman Minoonejad, Seyed Hamed Mousavi, Hamed Abbasi.

**Investigation:** Seyed Hamed Mousavi.

**Methodology:** Saeedeh Abbasi, Hooman Minoonejad, Seyed Hamed Mousavi.

**Project administration:** Saeedeh Abbasi, Hooman Minoonejad.

**Resources:** Saeedeh Abbasi, Seyed Hamed Mousavi.

**Software:** Saeedeh Abbasi.

**Supervision:** Saeedeh Abbasi, Hooman Minoonejad.

**Validation:** Hooman Minoonejad, Hamed Abbasi.

**Writing – original draft:** Saeedeh Abbasi.

**Writing – review & editing:** Hooman Minoonejad, Seyed Hamed Mousavi, Hamed Abbasi.

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
