## [Decision Letter · Decision Letter 0]

22 Nov 2024

PONE-D-24-46526The Impact of different Standing positions on Pelvic Muscle Activation and Lumbar Lordosis in LBP-developers during prolonged standingPLOS ONE

Dear Dr. Abbasi,

Thank you for submitting your manuscript to PLOS ONE. After careful consideration, we feel that it has merit but does not fully meet PLOS ONE’s publication criteria as it currently stands. Therefore, we invite you to submit a revised version of the manuscript that addresses the points raised during the review process.

We look forward to receiving your revised manuscript.

Kind regards,

Yaodong Gu

Academic Editor

PLOS ONE

Journal Requirements:

Reviewers' comments:

Reviewer's Responses to Questions

**Comments to the Author**

1. Is the manuscript technically sound, and do the data support the conclusions?

Reviewer #1: Yes

Reviewer #2: Partly

2. Has the statistical analysis been performed appropriately and rigorously? 

Reviewer #1: Yes

Reviewer #2: Yes

3. Have the authors made all data underlying the findings in their manuscript fully available?

Reviewer #1: Yes

Reviewer #2: No

4. Is the manuscript presented in an intelligible fashion and written in standard English?

Reviewer #1: Yes

Reviewer #2: Yes

5. Review Comments to the Author

Reviewer #1: Reviewer 1

Comments and Suggestions for Authors:

1) The current introduction section conflates multiple factors, resulting in a slightly weaker logical context for the study. The description of non-critical factors (e.g., excessive VAS discussion) should be simplified to focus more on the characteristics of the PD population versus the core issue of standing posture interventions.

2) Lack of background on some of the core literature, especially the key studies on the relationship between lumbar lordosis angle and LBP, did not provide enough detailed discussion.

3) The current introduction is not specific enough in describing the research gaps and fails to clearly state the specific areas where the current literature is lacking. It is recommended that the innovations of the study be more clearly articulated.

4) Lines 57-59 refer to “task type” affecting Gmed co-activation, but do not account for its direct association with PD or LBP.

5) The objectives of the study in lines 91-94 should be more specific and should make a natural transition with the logic of the preceding text to avoid abruptness.

6) The description of “suitable height condition” in line 84 is not supported by clear data and it is recommended that the appropriate experimental background be added.

7) No mention was made of adjusting for confounding variables (e.g., height, weight, etc.) or stratifying the analysis. In addition, the ANOVA model lacked explicit assumptions and explanations, although it mentioned time and condition interactions.

8) Some of the sentences are lengthy and contain grammatical errors (e.g., line 125, “participants ‘informed consent’ is not clear), and the descriptions are repetitive.

9) Lines 182: There is a lack of detailed description of MVC and the details of solving the entire muscle activation calculation process. Please add the relevant content, the authors may consider referring to: Accurately and effectively predict the ACL force: Utilizing biomechanical landing pattern before and after-fatigue (https://doi.org/10.1016/j.cmpb.2023.107761); New insights optimize landing strategies to reduce lower limb injury risk (https://doi.org/10.34133/cbsystems.0126)

10) The article has a relatively small set of indicators, and more data indicators could be added to better explore the research questions.

Reviewer #2: The Impact of different Standing positions on Pelvic Muscle Activation and Lumbar

Lordosis in LBP-developers during prolonged standing

Congratulations it was a good manuscript. But there are several critical points that need to be made clear. You can see it in the attached file.

6. PLOS authors have the option to publish the peer review history of their article (what does this mean? ). If published, this will include your full peer review and any attached files.

**Do you want your identity to be public for this peer review?** For information about this choice, including consent withdrawal, please see our Privacy Policy .

Reviewer #1: No

Reviewer #2: **Yes: ** Mohamadreza Hatefi

---

## [Author Response · Author response to Decision Letter 1]

3 Dec 2024

Dear editor and reviewers;

We thank you for your insightful and constructive suggestions. Based on your helpful suggestions, we were able to further improve our manuscript. We carefully considered and addressed all your specific comments and revised the text if necessary.

Please find your comments in bold and our responses in italic font. In the manuscript, the highlighted parts in green are the revisions made based on the comments of reviewer1 and the highlighted parts in yellow are the revisions made based on the comments of reviewer2.

Thank you for taking the time to provide detailed feedback on our manuscript. We appreciate your insights and constructive criticism. Your comments will be invaluable in refining our study. We carefully address each of the points you have raised to enhance the clarity of our rationale, align the benefits with the actual outcomes, improve the accuracy of our results, and enhance the structure of our discussion. Your input will undoubtedly contribute to the overall quality of our work.

Reviewer #1

Introduction

1* The current introduction section conflates multiple factors, resulting in a slightly weaker logical context for the study. The description of non-critical factors (e.g., excessive VAS discussion) should be simplified to focus more on the characteristics of the PD population versus the core issue of standing posture interventions.

Thank you for your comment. We added dedicated paragraphs discussing the critical role of Gmed muscle co-activation, lumbar lordosis, footrest, and arm position during standing, highlighting its relevance to PDs and its implications for LBP development.

2*Lack of background on some of the core literature, especially the key studies on the relationship between lumbar lordosis angle and LBP, did not provide enough detailed discussion.

Thank you for your comment. We expanded the discussion on the relationship between lumbar lordosis and LBP, referencing specific studies and linking these findings to PD characteristics and interventions:

Lumbar lordosis is another significant factor in LBP development during prolonged standing. Research indicates that PDs exhibit significantly greater lumbar lordosis than non-PDs, with a strong correlation between lordosis severity and LBP prevalence [31,32]. During prolonged standing, spinal posture may interact with the stresses placed on the spine, increasing the risk of developing LBP symptoms. These findings are further supported by the significant positive relationship between symptom severity and the degree of lordosis observed, where greater lordosis was associated with greater severity of LBP symptoms, even with acute and transient symptoms during the standing test [31].

3*The current introduction is not specific enough in describing the research gaps and fails to clearly state the specific areas where the current literature is lacking. It is recommended that the innovations of the study be more clearly articulated.

Thank you for your comment. We Clearly articulated the gaps in the literature, emphasizing the lack of studies on combined interventions (e.g., footrest height and arm positions) for alleviating pain in PDs:

Since lumbar lordosis plays a pivotal role in the development of LBP, and prior research highlights the significant impact of arm positioning on lordosis, the lack of studies addressing this relationship in PDs constitutes a notable research gap. Moreover, footrest interventions have been shown to effectively reduce the activity of the Gmed and lumbar muscles in PDs. This study investigates the combined effects of arm positioning and footrest usage during one hour of standing to improve lumbar-pelvic stability, alleviate pain intensity, and offer practical, generalizable solutions for reducing LBP risks in occupations requiring prolonged standing.

4* Lines 57-59 refer to “task type” affecting Gmed co-activation, but do not account for its direct association with PD or LBP.

Thank you for your notice. We expanded the discussion to explicitly link task type to Gmed muscle co-activation and its potential role in LBP and PDs, as highlighted in relevant studies:

Notably, task type during standing directly influences Gmed co-activation and may affect PDs' susceptibility to LBP. For example, tasks involving upper limb activity, such as assembly or sorting, tend to increase Gmed co-activation, while tasks without upper limb movement reduce it [30]. This association highlights the importance of investigating how task-specific factors influence muscle activation patterns in PDs. This association is important because increased Gmed co-activation helps maintain lumbar stability and prevent LBP. Inadequate or inconsistent Gmed activation, especially during dynamic tasks, may increase vulnerability to LBP in PDs.

5* The objectives of the study in lines 91-94 should be more specific and should make a natural transition with the logic of the preceding text to avoid abruptness.

Thank you for your notice. We refined the study objectives for specificity and ensured they follow logically from the preceding discussion :

Since lumbar lordosis plays a pivotal role in the development of LBP, and prior research highlights the significant impact of arm positioning on lordosis, the lack of studies addressing this relationship in PDs constitutes a notable research gap. Moreover, footrest interventions have been shown to effectively reduce the activity of the Gmed and lumbar muscles in PDs. This study investigates the combined effects of arm positioning and footrest usage during one hour of standing to improve lumbar-pelvic stability, alleviate pain intensity, and offer practical, generalizable solutions for reducing LBP risks in occupations requiring prolonged standing.

6* The description of “suitable height condition” in line 84 is not supported by clear data and it is recommended that the appropriate experimental background be added.

Thank you for your notice. We added supporting experimental evidence from cited studies :

So, adjusting footrest height and usage duration is crucial for achieving better outcomes [16,47]. A footrest height equivalent to 10% of body height or adjusted to achieve a 135-degree trunk-to-thigh angle has been shown to reduce lumbar load, fatigue, and pain intensity [44,45]. These devices, combined with optimized upper limb positions, may improve lumbopelvic alignment and alleviate discomfort during prolonged standing.

Method

7* Some of the sentences are lengthy and contain grammatical errors (e.g., line 125, “participants ‘informed consent’ is not clear), and the descriptions are repetitive.

Thank you for your notice. I refined sentences and clarified but the excessive data in this paragraph refer to the editor’s comments that were added, so I cannot remove some of this information:

Informed consent was obtained from participants for their involvement in the study and for the publication of identifiable information or images in an open-access online journal. During data collection, access to personally identifiable information was restricted based on this consent, and all data were anonymized for analysis.

8* No mention was made of adjusting for confounding variables (e.g., height, weight, etc.) or stratifying the analysis. In addition, the ANOVA model lacked explicit assumptions and explanations, although it mentioned time and condition interactions.

Thank you for your notice. Participants with a body mass index (BMI) greater than 30 were excluded. The BMI range of the included participants was between 18 and 22, and their heights were also within a similar range. Therefore, due to the homogeneity of these variables within the sample, their potential influence as confounding variables was negligible. This matter refined in manuscript as bellow:

Also ,participants were excluded if they had employment involving standing in one place for more than 1 hour per day in the past 12 months, were unable to stand for more than 4 hours, had a body mass index (BMI) greater than 30, or reported any symptoms of LBP at the start of the standing task.

Data Analysis

9* Lines 182: There is a lack of detailed description of MVC and the details of solving the entire muscle activation calculation process. Please add the relevant content, the authors may consider referring to: Accurately and effectively predict the ACL force: Utilizing biomechanical landing pattern before and after-fatigue (https://doi.org/10.1016/j.cmpb.2023.107761); New insights optimize landing strategies to reduce lower limb injury risk (https://doi.org/10.34133/cbsystems.0126)

Thank you for your valuable comment. We have revised the manuscript to include a detailed description of the MVC normalization process and the muscle activation calculation. Specifically, we have added the following details:

• The procedure for obtaining MVC values through maximum isometric contractions, including the duration and method of calculation.

• The normalization of EMG signals to %MVC for consistency and comparability across participants and conditions.

• A clear explanation of the calculation pipeline, including the use of RMS for filtered EMG data and further analysis conducted in MATLAB.

For normalization, the electromyographic signals were processed by subtracting resting activation levels and then normalizing to maximum voluntary contraction (MVC). MVC values were obtained by asking participants to perform maximum isometric contractions against resistance for each targeted muscle, following standard testing procedures as described in [reference specific source]. These contractions were held for 5 seconds, with the middle 3 seconds of the contraction used to compute the MVC[24]. The root mean square (RMS) of the filtered raw data was then calculated for subsequent analysis. This normalization ensured that the muscle activation data were expressed as a percentage of MVC (%MVC), allowing for comparisons across participants and conditions. Raw EMG data were imported directly into a custom MATLAB (V.4) script for further processing, which included computing the time-domain and frequency-domain characteristics of the muscle activation.

10* The article has a relatively small set of indicators, and more data indicators could be added to better explore the research questions.

We appreciate your insightful comment. While our study aimed to prioritize key indicators that are most directly relevant to the research questions, we recognize that adding more indicators could provide deeper insights. We will take this into account in designing future studies and expanding the scope of our analysis

Reviewer #2

Title

1* You have just investigated the activation of the gluteus medius! Therefore, gluteus medius activation should be used instead of pelvic muscle activation.

Thank you for your notice. The title has been updated accordingly, as shown below:

The Impact of different Standing positions on Gluteus Medius Activation and Lumbar Lordosis in LBP-developers during prolonged standing

Abstract

2* L26: Use gluteus “medius muscle activity” instead of “muscle activity” in the all manuscript.

Thank you for your notice. It has been corrected throughout the entire manuscript to "gluteus medius muscle activity."

3* P2: Report exact p-value separately for significant parameters in results section.

Thank you for your comment. Because the dataset includes numerous parameters for time, condition, and their interaction (time * condition) across three variables and two groups, only the most important results are reported and we added as bellow:

No significant interaction was observed between the groups in right and left Gmed co-contraction index (CCI) (p = 0.14). However, both groups exhibited a significant decrease in CCI during prolonged standing (time * condition: p = 0.003). Additionally, Group B consistently demonstrated lower overall levels of co-contraction across time (p = 0.01). An approximate 6-degree reduction in lumbar lordosis was observed after prolonged standing with both interventions (group A and group B; p = 0.008 and p = 0.01, respectively), although no significant differences in lumbar lordosis were detected between the groups. Lumbar discomfort increased over time; however, the interventions significantly alleviated this discomfort after the intervention time point. Notably, group B reported lower pain intensity compared to group A (p = 0.007).

4* P2: You must also specify the type of study as well as the statistical test used.

Thank you for your notice. We added this sentences as below:

These variables were analyzed using repeated measures (between/within) ANOVA.

Introduction

5* P3: The bilateral co-activation of Gmed was one of the dependent variables in your study, but you did not mention any related articles to indicate its importance in PD or LBP.

Thank you for your notice. We added a dedicated paragraph discussing the critical role of Gmed muscle co-activation in lumbar and pelvic stability during standing, highlighting its relevance to PDs and its implications for LBP development:

The bilateral co-activation of the gluteus medius (Gmed) muscle plays a critical role in maintaining pelvic and lumbar stability during standing. Impaired or altered Gmed activation has been linked to LBP in PDs, contributing to changes in motor control and muscle co-activation [8,12]. It is believed that this co-activation compensates for the inability to properly control the trunk by the core muscles for postural stability [9,26,27]. Additionally, the low strength and endurance of this muscle are significantly associated with increased co-activation during prolonged standing [28].

6* L 87: What did you mean by “smaller anterior-posterior height ratio”? It is unclear!

Thank you for your comment. We rephrased this to clearly describe:

Women generally exhibit greater lumbar flexion (a smaller anterior-posterior height ratio of intervertebral discs) compared to men, who demonstrate greater lumbar extension (a larger ratio).

Methods

7* P6: Please explain how did you estimate the sample size?

Thank you for your comment. We clarified as bellow:

The sample size was determined using G*Power software version 3.1. With a confidence level of 0.95, an effect size of 0.83[24], and a test power of 0.85, the minimum required sample size was calculated to be 8 participants. However, considering that previous studies included at least 12 participants, the test groups were set to 12 participants each (a total of 24 participants).

8* P6: You mentioned that there were two groups; however, there is no information provided regarding the demographics of each group separately or a comparison between them. Additionally, please clarify the randomization methods used.

Thank you for your valuable feedback. Both groups were entirely similar in terms of demographics, including age, gender, height, and weight, as all participants were PD.

Participants were randomly assigned to each group using "a computer-generated randomization list”.

9* L108: Please clarify the time of the “prolonged standing”!

Thank you for your comment. We clarified as bellow:

In the initial step, all participants underwent a 2-hours prolonged standing test to categorize them as either PD or NPD.

On a separate day, participants stood for one hour in a confined workspace.

10* P 9: Why didn’t you consider the subject's lower extremity length or total height to determine the box height? This difference in subjects' anthropometric properties obviously affects lower extremity angles and muscle activity!

Thank you for your comment. The mean height is : 164.35 (SD: 4.76) cm . We clarified as bellow:

A 20-cm footrest was used during the standing period. This height was selected to represent 10% of the participants' average body height and to ensure a 135° trunk-to-thigh angle [44,45] for the majority of participants, which ranged between 17 and 20 cm.

11* P 9: It is unclear what the duration of the Gmed muscle activity record is. Have you recorded the muscle activation in which position, and for how many seconds?

Thank you for your comment. We clarified as bellow:

For each five-minute trial, data were segmented into three one-minute windows representi

---

## [Decision Letter · Decision Letter 1]

5 Dec 2024

PONE-D-24-46526R1The Impact of different Standing positions on Gluteus Medius Activation and Lumbar Lordosis in LBP-developers during prolonged standingPLOS ONE

Dear Dr. Minoonejad,

Thank you for submitting your manuscript to PLOS ONE. After careful consideration, we feel that it has merit but does not fully meet PLOS ONE’s publication criteria as it currently stands. Therefore, we invite you to submit a revised version of the manuscript that addresses the points raised during the review process.

We look forward to receiving your revised manuscript.

Kind regards,

Yaodong Gu

Academic Editor

PLOS ONE

Reviewers' comments:

Reviewer's Responses to Questions

**Comments to the Author**

1. If the authors have adequately addressed your comments raised in a previous round of review and you feel that this manuscript is now acceptable for publication, you may indicate that here to bypass the “Comments to the Author” section, enter your conflict of interest statement in the “Confidential to Editor” section, and submit your "Accept" recommendation.

Reviewer #1: (No Response)

Reviewer #2: All comments have been addressed

2. Is the manuscript technically sound, and do the data support the conclusions?

Reviewer #1: Yes

Reviewer #2: Yes

3. Has the statistical analysis been performed appropriately and rigorously? 

Reviewer #1: Yes

Reviewer #2: Yes

4. Have the authors made all data underlying the findings in their manuscript fully available?

Reviewer #1: Yes

Reviewer #2: Yes

5. Is the manuscript presented in an intelligible fashion and written in standard English?

Reviewer #1: Yes

Reviewer #2: Yes

6. Review Comments to the Author

Reviewer #1: After an in-depth review of the author's revisions to the manuscript, I appreciate the efforts and improvements made by the author. However, to further enhance the depth and breadth of the study, I suggest that the authors make secondary revisions in the following more challenging aspects:

1. The authors mentioned the relationship between low back pain (LBP) and prolonged standing and specific job tasks in the context of the study, but the current literature review seems to be insufficient. It is suggested that the author further explore the latest research progress on the pathogenesis of LBP in recent years, especially the cutting-edge theories related to muscle activation pattern, lumbar stability and task specificity. This helps to provide the reader with richer background information and strengthens the theoretical support of research.

2. In describing the research method, although the author mentioned the calculation of the MVC and the normalization processing of muscle activation data, the detailed implementation details of these steps and the verification process were not detailed enough. It is recommended that the author add detailed steps about the MVC measurement method, including the equipment used, the measurement position, the posture requirements when measuring, etc., and explain why this method was chosen for MVC measurement. At the same time, for the normalization of muscle activation data, the author should provide detailed instructions on how to screen and filter the original data, how to calculate the root mean square value (RMS) and how to perform the normalization processing and attach the relevant formulas to increase the transparency and repeatability of the study.

3. When presenting the results of the study, the author mainly described the changes in muscle activation patterns under different conditions. However, the authors' analysis of the mechanisms behind these changes does not seem to go far enough. It is suggested that the authors combine the biomechanical and physiological principles to further explore how these changes affect the stability of lumbar spine and the mechanism of LBP. For example, it is possible to analyze the relationship between changes in muscle activation patterns in different work tasks and lumbar load, and how these changes affect the stress distribution and stability of the lumbar spine.

4. The author summarizes the research findings in conclusion but mentions the future research direction in a relatively general way. It is suggested that the authors propose more specific and in-depth future research topics based on the research results. For example, the risk of LBP could be further reduced by optimizing job task design, improving the work environment, or introducing specific rehabilitation training methods. At the same time, the authors may also consider combining the results of this study with research in other related fields to form a more comprehensive research perspective.

5. Although the author has already mentioned some limitations in the study, I think the discussion on methodological limitations could go further. For example, authors may explore the impact of sample size, study design (such as the lack of randomized controlled trials), measurement errors, and other factors on study results, and suggest possible solutions or improvements. This helps readers gain a more complete understanding of the limitations of the study and stimulates more thinking about how to improve the research methodology.

It is hoped that the author can make two modifications to the above suggestions to further enhance the depth and breadth of the study.

Reviewer #2: The authors addressed my all comments: The Impact of different Standing positions on Gluteus Medius Activation and Lumbar Lordosis in LBP-developers during prolonged standing

7. PLOS authors have the option to publish the peer review history of their article (what does this mean? ). If published, this will include your full peer review and any attached files.

**Do you want your identity to be public for this peer review?** For information about this choice, including consent withdrawal, please see our Privacy Policy .

Reviewer #1: No

Reviewer #2: **Yes: ** Mohamadreza Hatefi

---

## [Author Response · Author response to Decision Letter 2]

16 Dec 2024

Dear editor and reviewers;

We thank you for your insightful and constructive suggestions. Based on your helpful suggestions, we were able to further improve our manuscript. We carefully considered and addressed all your specific comments and revised the text if necessary.

Please find your comments in bold and our responses in italic font. In the manuscript, the highlighted parts in yellow are the revisions made based on the comments of reviewer1 .

Reviewer #1

1. The authors mentioned the relationship between low back pain (LBP) and prolonged standing and specific job tasks in the context of the study, but the current literature review seems to be insufficient. It is suggested that the author further explore the latest research progress on the pathogenesis of LBP in recent years, especially the cutting-edge theories related to muscle activation pattern, lumbar stability and task specificity. This helps to provide the reader with richer background information and strengthens the theoretical support of research.

Thank you for your comment. We revised the introduction paragraphs as below:

The etiology of LBP involves complex interactions between various factors, including anthropometric, structural, psychological, postural, and muscular aspects [6,10–17]. Reports suggest that individuals prone to standing-induced LBP, or Pain Developers (PDs), frequently exhibit biomechanical and motor control deficits. For instance, PDs demonstrate lower Gluteus Medius (GMed) endurance during side-bridge tests, often accompanied by reduced strength [5,18,19]. The bilateral co-contraction of Gmed muscle plays a critical role in maintaining pelvic and lumbar stability during standing. Impaired or altered Gmed activation has been linked to LBP in PDs, contributing to changes in motor control and muscle co-contraction [20,21]. It is believed that this co-contraction compensates for the inability to properly control the trunk by the core muscles for postural stability [12,22,23]. The active hip abduction (AHAbd) test is often used to identify these deficits, as it predicts standing-induced LBP with high accuracy (0.83) [21,24,25]. Furthermore, task-specific factors, such as upper limb activity during standing, significantly influence Gmed co-contraction, with implications for LBP development. Tasks requiring upper limb movement, like sorting or assembly, change Gmed co-contraction, potentially reducing trunk instability and LBP risks [26]. Inadequate or inconsistent GMed activation, especially during dynamic tasks, may increase vulnerability to LBP in PDs. Lumbar lordosis is another pivotal factor in LBP development. PDs generally exhibit greater lumbar lordosis compared to non-PDs, with higher lordosis severity correlating with increased LBP symptoms [27,28]. One study noted a significant correlation with maximal visual analog scale (VAS) scores [28]. This association underscores the interplay between spinal posture and mechanical stress during prolonged standing. Research further links lumbar lordosis severity to negative sagittal vertical axis (SVA) displacement, indicating that postural deviations may exacerbate LBP symptoms [28–30]. Gender differences further complicate the understanding of LBP in PDs. Women generally exhibit greater lumbar flexion (a smaller anterior-posterior height ratio of intervertebral discs) compared to men. Women also tend to have higher endurance but reduced force capacity, suggesting gender-specific contributions to LBP susceptibility and motor control adaptations [31,32].

2. In describing the research method, although the author mentioned the calculation of the MVC and the normalization processing of muscle activation data, the detailed implementation details of these steps and the verification process were not detailed enough. It is recommended that the author add detailed steps about the MVC measurement method, including the equipment used, the measurement position, the posture requirements when measuring, etc., and explain why this method was chosen for MVC measurement. At the same time, for the normalization of muscle activation data, the author should provide detailed instructions on how to screen and filter the original data, how to calculate the root mean square value (RMS) and how to perform the normalization processing and attach the relevant formulas to increase the transparency and repeatability of the study.

We appreciate your suggestion. We expanded the method as below:

Maximum voluntary contraction (MVC) values were measured by instructing participants to perform maximal isometric contractions against resistance. To normalize and compare muscle activity relative to each muscle's maximal strength, MVCs were recorded for the left and right gluteus medius (Gmed). Participants performed hip abduction in a side-lying position, with resistance applied at the ankle by the experimenter, to obtain MVCs [55,56]. The experimenter can actively monitor and adjust the resistance applied, ensuring participants are performing maximal efforts without exceeding their limits, which could lead to injury. Each leg underwent two sets of MVCs, with a minimum of 30 seconds of rest between efforts. Additionally, 10-second resting trials were conducted with participants lying in both the supine and prone positions [3].

Each EMG linear envelope was normalized as a percentage of the maximum voluntary contraction (MVC). Subsequently, the root mean square (RMS) of the filtered raw data was calculated to determine the average intensity of the signal. This was achieved by averaging the squared values using the formula 1/N∑ xi2. The normalized muscle activation data were expressed as a percentage of MVC using the formula (RMS EMG/ MVC RMS)*100. This normalization facilitated comparisons across participants and conditions [31].

3. When presenting the results of the study, the author mainly described the changes in muscle activation patterns under different conditions. However, the authors' analysis of the mechanisms behind these changes does not seem to go far enough. It is suggested that the authors combine the biomechanical and physiological principles to further explore how these changes affect the stability of lumbar spine and the mechanism of LBP. For example, it is possible to analyze the relationship between changes in muscle activation patterns in different work tasks and lumbar load, and how these changes affect the stress distribution and stability of the lumbar spine.

Thank you for pointing this out. We revised the discussion according to your comment as shown in the highlighted parts:

The arm position alteration, involving shoulder flexion with hands crossed on the clavicles may influence upper body muscle activation, indirectly affecting lumbar alignment through kinetic chain interactions. Given that the weight of the arms increases the torque on the spine when moving forward and requires increased back muscle strength to maintain balance, this increase in muscle strength affects the fluctuation of body posture and consequently changes lumbo-pelvic parameters, including the amount of activity of the trunk muscles in stabilizing the back during standing [33]. Also, using footrest can cause slight lumbar flexion, reduced lumbar torque, and decreased lumbar erector spinae muscle activity [42].

These findings suggest that footrest use and arm position adjustments optimize biomechanical alignment, alleviate lumbar strain, and enhance postural stability during prolonged standing. Moreover, engaging the upper extremities through core muscle activation can reduce Gmed's compensatory role in trunk control and postural stability [12,22,23]. This reduction in Gmed muscle strain during intermittent foot elevation and changing arm position may serve to alleviate some of the discomfort LBP.

It is conceivable that altering arm positioning helps decrease lumbar extension demands, allowing for reduced lumbar muscle activation and alleviating muscle fatigue [38,59]. This adjustment not only contributes to pain relief but also underscores the role of strategic muscle activation as a stabilizing mechanism for prolonged standing and reducing the risk of lumbar instability and LBP.

4. The author summarizes the research findings in conclusion but mentions the future research direction in a relatively general way. It is suggested that the authors propose more specific and in-depth future research topics based on the research results. For example, the risk of LBP could be further reduced by optimizing job task design, improving the work environment, or introducing specific rehabilitation training methods. At the same time, the authors may also consider combining the results of this study with research in other related fields to form a more comprehensive research perspective.

Thank you for your notice. We expanded the conclusion as below:

These findings have practical implications for designing workplace interventions to reduce musculoskeletal strain and improve comfort for individuals required to stand for extended periods in various occupational settings. One potential avenue is improving the work environment—such as adjusting workstation height, providing adjustable footrests, or ensuring proper ergonomic setups—could offer further relief by promoting better spinal alignment and reducing muscle fatigue. Future research should explore the long-term effects of these interventions and investigate other potential combinations of more movements and positions to further enhance their effectiveness.

5. Although the author has already mentioned some limitations in the study, I think the discussion on methodological limitations could go further. For example, authors may explore the impact of sample size, study design (such as the lack of randomized controlled trials), measurement errors, and other factors on study results, and suggest possible solutions or improvements. This helps readers gain a more complete understanding of the limitations of the study and stimulates more thinking about how to improve the research methodology.

Thank you for pointing this out. We refined as below:

While the study provides compelling preliminary evidence, several methodological limitations need to be considered. One of the main limitations is the small sample size, which limits the generalizability of the findings. A larger, more diverse sample would allow for better representation of various populations, particularly those with occupational standing requirements. Moreover, the lack of long-term follow-up limits the ability to assess the sustained effects of the interventions. Future studies should include longitudinal designs to examine whether the benefits observed in this study can be maintained over time and how they influence long-term musculoskeletal health.

---

## [Decision Letter · Decision Letter 2]

26 Dec 2024

The Impact of different Standing positions on Gluteus Medius Activation and Lumbar Lordosis in LBP-developers during prolonged standing

PONE-D-24-46526R2

Dear Dr. Minoonejad,

We’re pleased to inform you that your manuscript has been judged scientifically suitable for publication and will be formally accepted for publication once it meets all outstanding technical requirements.

Kind regards,

Yaodong Gu

Academic Editor

PLOS ONE

Additional Editor Comments (optional):

Reviewers' comments:

Reviewer's Responses to Questions

**Comments to the Author**

1. If the authors have adequately addressed your comments raised in a previous round of review and you feel that this manuscript is now acceptable for publication, you may indicate that here to bypass the “Comments to the Author” section, enter your conflict of interest statement in the “Confidential to Editor” section, and submit your "Accept" recommendation.

Reviewer #1: All comments have been addressed

Reviewer #2: All comments have been addressed

2. Is the manuscript technically sound, and do the data support the conclusions?

Reviewer #1: Yes

Reviewer #2: Yes

3. Has the statistical analysis been performed appropriately and rigorously? 

Reviewer #1: Yes

Reviewer #2: Yes

4. Have the authors made all data underlying the findings in their manuscript fully available?

Reviewer #1: Yes

Reviewer #2: Yes

5. Is the manuscript presented in an intelligible fashion and written in standard English?

Reviewer #1: Yes

Reviewer #2: Yes

6. Review Comments to the Author

Reviewer #1: (No Response)

Reviewer #2: The authors investigated the Impact of different Standing positions on Gluteus Medius Activation and Lumbar Lordosis in LBP-developers during prolonged standing.

Based in the last revision, all comments have been addressed.

7. PLOS authors have the option to publish the peer review history of their article (what does this mean? ). If published, this will include your full peer review and any attached files.

**Do you want your identity to be public for this peer review?** For information about this choice, including consent withdrawal, please see our Privacy Policy .

Reviewer #1: No

Reviewer #2: **Yes: ** Mohamadreza Hatefi

---

## [Editor Report · Acceptance letter]

PONE-D-24-46526R2

PLOS ONE

Dear Dr. Minoonejad,

I'm pleased to inform you that your manuscript has been deemed suitable for publication in PLOS ONE. Congratulations! Your manuscript is now being handed over to our production team.

Kind regards,

on behalf of

Professor Yaodong Gu

Academic Editor

PLOS ONE